# DiffTaichi: Differentiable Programming for Physical Simulation

**Yuanming Hu**[†]**, Luke Anderson**[†]**, Tzu-Mao Li**[∗]**, Qi Sun**[‡]**, Nathan Carr**[‡]**,
Jonathan Ragan-Kelley**[∗]**, Frédo Durand**[†]

[†]MIT CSAIL      `{yuanming,lukea,fredo}@mit.edu`
[‡]Adobe Research    `{qisu,ncarr}@adobe.com`
[∗]UC Berkeley      `{tzumao,jrk}@berkeley.edu`

## Abstract

We present DiffTaichi, a new differentiable programming language tailored for building high-performance differentiable physical simulators. Based on an imperative programming language, DiffTaichi generates gradients of simulation steps using source code transformations that preserve arithmetic intensity and parallelism. A light-weight tape is used to record the whole simulation program structure and replay the gradient kernels in a reversed order, for end-to-end backpropagation. We demonstrate the performance and productivity of our language in gradient-based learning and optimization tasks on 10 different physical simulators. For example, a differentiable elastic object simulator written in our language is $4.2\times$ shorter than the hand-engineered CUDA version yet runs as fast, and is $188\times$ faster than the TensorFlow implementation. Using our differentiable programs, neural network controllers are typically optimized within only tens of iterations.

## 1 Introduction

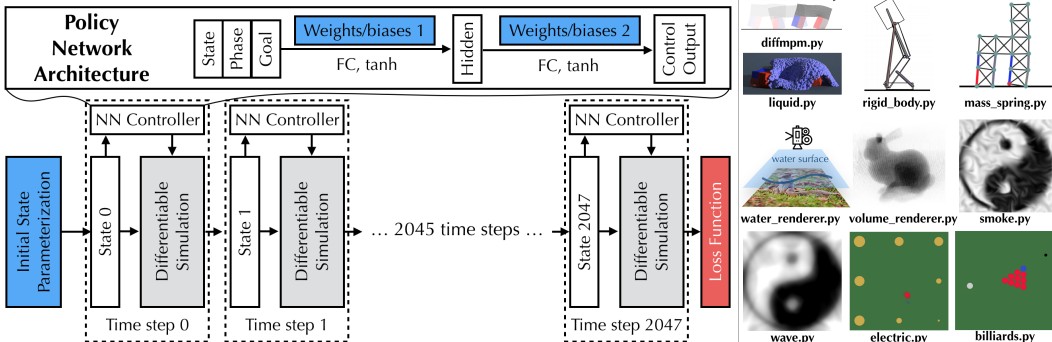

Figure 1: **Left:** Our language allows us to seamlessly integrate a neural network (NN) controller and a physical simulation module, and update the weights of the controller or the initial state parameterization (blue). Our simulations typically have $512 \sim 2048$ time steps, and each time step has up to one thousand parallel operations. **Right:** 10 differentiable simulators built with DiffTaichi.

Differentiable physical simulators are effective components in machine learning systems. For example, de Avila Belbute-Peres et al. (2018a) and Hu et al. (2019b) have shown that controller optimization with differentiable simulators converges one to four orders of magnitude faster than model-free reinforcement learning algorithms. The presence of differentiable physical simulators in the inner loop of these applications makes their performance vitally important. Unfortunately, using existing tools it is difficult to implement these simulators with high performance.

We present *DiffTaichi*, a new differentiable programming language for high performance physical simulations on both CPU and GPU. It is based on the Taichi programming language (Hu et al.,

2019a). The DiffTaichi automatic differentiation system is designed to suit key language features required by physical simulation, yet often missing in existing differentiable programming tools, as detailed below:

**Megakernels**    Our language uses a "megakernel" approach, allowing the programmer to naturally fuse multiple stages of computation into a single kernel, which is later differentiated using source code transformations and just-in-time compilation. Compared to the linear algebra operators in TensorFlow (Abadi et al., 2016) and PyTorch (Paszke et al., 2017), DiffTaichi kernels have higher arithmetic intensity and are therefore more efficient for physical simulation tasks.

**Imperative Parallel Programming**    In contrast to functional array programming languages that are popular in modern deep learning (Bergstra et al., 2010; Abadi et al., 2016; Li et al., 2018b), most traditional physical simulation programs are written in imperative languages such as Fortran and C++. DiffTaichi likewise adopts an imperative approach. The language provides parallel loops and control flows (such as "if" statements), which are widely used constructs in physical simulations: they simplify common tasks such as handling collisions, evaluating boundary conditions, and building iterative solvers. Using an imperative style makes it easier to port existing physical simulation code to DiffTaichi.

**Flexible Indexing**    Existing parallel differentiable programming systems provide element-wise operations on arrays of the same shape, e.g. `c[i, j] = a[i, j] + b[i, j]`. However, many physical simulation operations, such as numerical stencils and particle-grid interactions are not element-wise. Common simulation patterns such as `y[p[i] * 2, j] = x[q[i + j]]` can only be expressed with unintuitive `scatter`/`gather` operations in these existing systems, which are not only inefficient but also hard to develop and maintain. On the other hand, in DiffTaichi, the programmer directly manipulates array elements via arbitrary indexing, thus allowing partial updates of global arrays and making these common simulation patterns naturally expressible. The explicit indexing syntax also makes it easy for the compiler to perform access optimizations (Hu et al., 2019a).

The three requirements motivated us to design a tailored two-scale automatic differentiation system, which makes DiffTaichi especially suitable for developing complex and high-performance differentiable physical simulators, possibly with neural network controllers (Fig. 1, left). Using our language, we are able to quickly implement and automatically differentiate 10 physical simulators[1], covering rigid bodies, deformable objects, and fluids (Fig. 1, right). A comprehensive comparison between DiffTaichiand other differentiable programming tools is in Appendix A.

## 2 BACKGROUND: THE TAICHI PROGRAMMING LANGUAGE

DiffTaichi is based on the Taichi programming language (Hu et al., 2019a). Taichi is an imperative programming language embedded in C++14. It delivers both high performance and high productivity on modern hardware. The key design that distinguishes Taichi from other imperative programming languages such as C++/CUDA is the decoupling of computation from data structures. This allows programmers to easily switch between different data layouts and access data structures with indices (i.e. `x[i, j, k]`), as if they are normal dense arrays, regardless of the underlying layout. The Taichi compiler then takes both the data structure and algorithm information to apply performance optimizations. Taichi provides "parallel-for" loops as a first-class construct. These designs make Taichi especially suitable for writing high-performance physical simulators. For more details, readers are referred to Hu et al. (2019a).

The DiffTaichi language frontend is embedded in Python, and a Python AST transformer compiles DiffTaichi code to Taichi intermediate representation (IR). Unlike Python, the DiffTaichi language is compiled, statically-typed, parallel, and differentiable. We extend the Taichi compiler to further compile and automatically differentiate the generated Taichi IR into forward and backward executables.

---

[1]Our language, compiler, and simulator code is open-source. All the results in this work can be reproduced by a single Python script. Visual results in this work are presented in the supplemental video.

We demonstrate the language using a mass-spring simulator, with three springs and three mass points, as shown right. In this section we introduce the forward simulator using the DiffTaichi frontend of Taichi, which is an easier-to-use wrapper of the Taichi C++14 frontend. 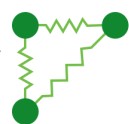

**Allocating Global Variables** Firstly we allocate a set of global tensors to store the simulation state. These tensors include a scalar `loss` of type `float32`, 2D tensors `x`, `v`, `force` of size `steps` ×`n_springs` and type `float32x2`, and 1D arrays of size `n_spring` for spring properties: `spring_anchor_a` `(int32)`, `spring_anchor_b (int32)`, `spring_length (float32)`.

**Defining Kernels** A mass-spring system is modeled by Hooke's law $\mathbf{F} = k(\|\mathbf{x}_a - \mathbf{x}_b\|_2 - l_0)\frac{\mathbf{x}_a - \mathbf{x}_b}{\|\mathbf{x}_a - \mathbf{x}_b\|_2}$ where $k$ is the spring stiffness, $\mathbf{F}$ is spring force, $\mathbf{x}_a$ and $\mathbf{x}_b$ are the positions of two mass points, and $l_0$ is the rest length. The following kernel loops over all the springs and scatters forces to mass points:

```
@ti.kernel
def apply_spring_force(t: ti.i32):
  # Kernels can have parameters. Here t is a parameter with type int32.
  for i in range(n_springs): # A parallel for, preferably on GPU
    a, b = spring_anchor_a[i], spring_anchor_b[i]
    x_a, x_b = x[t - 1, a], x[t - 1, b]
    dist = x_a - x_b
    length = dist.norm() + 1e-4
    F = (length - spring_length[i]) * spring_stiffness * dist / length
    # Apply spring impulses to mass points.
    force[t, a] += -F # "+=" is atomic by default
    force[t, b] +=  F
```

For each particle $i$, we use semi-implicit Euler time integration with damping: $\boldsymbol{v}_{t,i} = e^{-\Delta t \alpha} \boldsymbol{v}_{t-1,i} + \frac{\Delta t}{m_i}\mathbf{F}_{t,i}, \mathbf{x}_{t,i} = \mathbf{x}_{t-1,i} + \Delta t \boldsymbol{v}_{t,i}$, where $\boldsymbol{v}_{t,i}, \mathbf{x}_{t,i}, m_i$ are the velocity, position and mass of particle $i$ at time step $t$, respectively. $\alpha$ is a damping factor. The kernel is as follows:

```
@ti.kernel
def time_integrate(t: ti.i32):
  for i in range(n_objects):
    s = math.exp(-dt * damping) # Compile-time evaluation since dt and damping are constants
    v[t, i] = s * v[t - 1, i] + dt * force[t, i] / mass # mass = 1 in this example
    x[t, i] = x[t - 1, i] + dt * v[t, i]
```

**Assembling the Forward Simulator** With these components, we define the forward time integration:

```
def forward():
  for t in range(1, steps):
    apply_spring_force(t)
    time_integrate(t)
```

# 3 AUTOMATICALLY DIFFERENTIATING PHYSICAL SIMULATORS IN TAICHI

The main goal of DiffTaichi's automatic differentiation (AD) system is to generate gradient simulators automatically with **minimal code changes** to the traditional forward simulators.

**Design Decision** Source Code Transformation (SCT) (Griewank & Walther, 2008) and Tracing (Wengert, 1964) are common choices when designing AD systems. In our setting, using SCT to differentiate a whole simulator with thousands of time steps, results in high performance yet poor flexibility and long compilation time. On the other hand, naively adopting tracing provides flexibility yet poor performance, since the "megakernel" structure is not preserved during backpropagation. To get both performance and flexibility, we developed a **two-scale** automatic differentiation system (Figure 2): we use SCT for differentiating within kernels, and use a light-weight tape that only stores function pointers and arguments for end-to-end simulation differentiation. The global tensors are natural checkpoints for gradient evaluation.

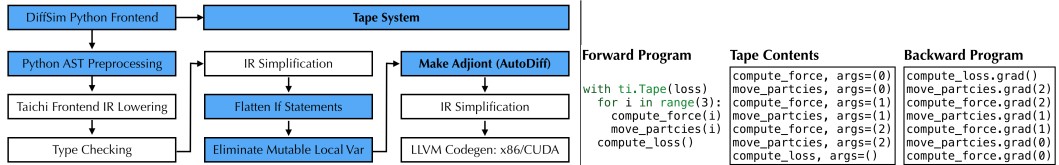

Figure 2: **Left:** The DiffTaichi system. We reuse some infrastructure (white boxes) from Taichi, while the blue boxes are our extensions for differentiable programming. **Right:** The tape records kernel launches and replays the gradient kernels in reverse order during backpropagation.

**Assumption** Unlike functional programming languages where immutable output buffers are generated, imperative programming allows programmers to freely modify global tensors. To make automatic differentiation well-defined under this setting, we make the following assumption on imperative kernels:

> **Global Data Access Rules:**
> 1) If a global tensor element is written more than once, then starting from the second write, the write must come in the form of an atomic add ("accumulation").
> 2) No read accesses happen to a global tensor element, until its accumulation is done.

In forward simulators, programmers may make subtle changes to satisfy the rules. For instance, in the mass-spring simulation example, we record the whole history of x and v, instead of keeping only the latest values. The memory consumption issues caused by this can be alleviated via checkpointing, as discussed later in Appendix D.

With these assumptions, kernels will not overwrite the outputs of each other, and the goal of AD is clear: given a primal kernel $f$ that takes as input $X_1, X_2, \ldots, X_n$ and outputs (or accumulates to) $Y_1, Y_2, \ldots, Y_m$, the generated gradient (adjoint) kernel $f^*$ should take as input $X_1, X_2, \ldots, X_n$ and $Y_1^*, Y_2^*, \ldots, Y_m^*$ and accumulate gradient contributions to $X_1^*, X_2^*, \ldots, X_m^*$, where each $X_i^*$ is an **adjoint** of $X_i$, i.e. $\partial(\text{loss})/\partial X_i$.

**Storage Control of Adjoint Tensors** Users can specify the storage of adjoint tensors using the Taichi data structure description language (Hu et al., 2019a), as if they are primal tensors. We also provide `ti.root.lazy_grad()` to automatically place the adjoint tensors following the layout of their primals.

### 3.1 LOCAL AD: DIFFERENTIATING TAICHI KERNELS USING SOURCE CODE TRANSFORMS

A typical Taichi kernel consists of multiple levels of for loops and a body block. To make later AD easier, we introduce two basic code transforms to simplify the loop body, as detailed below.

```
int a = 0;
if (b > 0) { a = b;}
  else     { a = 2b;}
a = a + 1;
return a;
```

```
// flatten branching
int a = 0;
a = select(b > 0, b, 2b);
a = a + 1
return a;
```

```
// eliminate mutable var

ssa1 = select(b > 0, b, 2b);
ssa2 = ssa1 + 1
return ssa2;
```

Figure 3: Simple IR preprocessing before running the AD source code transform (left to right). Demonstrated in c++. The actual Taichi IR is often more complex. Containing loops are ignored.

**Flatten Branching** In physical simulation branches are common, e.g., when implementing boundary conditions and collisions. To simplify the reverse-mode AD pass, we first replace "if" statements with ternary operators select(cond, value_if_true, value_if_false), whose gradients are clearly defined (Fig. 3, middle). This is a common transformation in program vectorization (e.g. Karrenberg & Hack (2011); Pharr & Mark (2012)).

**Eliminate Mutable Local Variables** After removing branching, we end up with straight-line loop bodies. To further simplify the IR and make the procedure truly single-assignment, we apply a series of local variable store forwarding transforms, until the mutable local variables can be fully eliminated (Fig. 3, right).

After these two custom IR simplification transforms, DiffTaichi only has to differentiate the straight-line code without mutable variables, which it achieves with reverse-mode AD, using a standard source code transformation (Griewank & Walther, 2008). More details on this transform are in Appendix B.

**Loops** Most loops in physical simulation are parallel loops, and during AD we preserve the parallel loop structures. For loops that are not explicitly marked as parallel, we reverse the loop order during AD transforms. We do not support loops that carry a mutating *local* variable since that would require a complex and costly run-time stack to maintain the history of local variables. Instead, users are instructed to employ *global* variables that satisfy the global data access rules.

**Parallelism and Thread Safety** For forward simulation, we inherit the "parallel-for" construct from Taichi to map each loop iteration onto CPU/GPU threads. Programmers use atomic operations for thread safety. Our system can automatically differentiate these atomic operations. Gradient contributions in backward kernels are accumulated to the adjoint tensors via atomic adds.

### 3.2 GLOBAL AD: END-TO-END BACKPROPAGATION USING A LIGHT-WEIGHT TAPE

We construct a tape (Fig. 2, right) of the kernel execution so that gradient kernels can be replayed in a reversed order. The tape is very light-weight: since the intermediate results are stored in global tensors, during forward simulation the tape only records kernel names and the (scalar) input parameters, unlike other differentiable functional array systems where all the intermediate buffers have to be recorded by the tape. Whenever a DiffTaichi kernel is launched, we append the kernel function pointer and parameters to the tape. When evaluating gradients, we traverse the reversed tape, and invoke the gradient kernels with the recorded parameters. Note that DiffTaichi AD is evaluating gradients with respect to input global tensors instead of the input parameters.

**Learning/Optimization with Gradients** Now we revisit the mass-spring example and make it differentiable for optimization. Suppose the goal is to optimize the rest lengths of the springs so that the triangle area formed by the three springs becomes 0.2 at the end of the simulation. We first define the loss function:

```python
@ti.kernel
def compute_loss(t: ti.i32):
  x01 = x[t, 0] - x[t, 1]
  x02 = x[t, 0] - x[t, 2]
  # Triangle area from cross product
  area = ti.abs(0.5 * (x01[0]*x02[1] - x01[1]*x02[0]))
  target_area = 0.2
  loss[None] = ti.sqr(area - target_area)
  # Everything in Taichi is a tensor.
  # "loss" is a scalar (0-D tensor), thereby indexed with [None].
```

**Goal:**
Adjust the spring rest lengths, so that this area=0.2 after 1024 time steps (initial area=0.005)

The programmer uses `ti.Tape` to memorize forward kernel launches. It automatically replays the gradients of these kernels in reverse for backpropagation. Initially the springs have lengths $[0.1, 0.1, 0.14]$, and after optimization the rest lengths are $[0.600, 0.600, 0.529]$. This means the springs will expand the triangle according to Hooke's law and form a larger triangle: [Reproduce: **mass_spring_simple.py**]

```
def main():
  for iter in range(200):
    with ti.Tape(loss):
      forward()
      compute_loss(steps - 1)
    print('Iter=', iter)
    print('Loss=',loss[None])
    # Gradient descent
    for i in range(n_springs):
      spring_length[i] -=
        lr * spring_length.grad[i]
```

Spring Rest Length Optimization

**Complex Kernels**   Sometimes the user may want to override the gradients provided by the compiler. For example, when differentiating a 3D singular value decomposition done with an iterative solver, it is better to use a manually engineered SVD derivative subroutine for better stability. We provide two more decorators `ti.complex_kernel` and `ti.complex_kernel_grad` to overwrite the default automatic differentiation, as detailed in Appendix C. Apart from custom gradients, complex kernels can also be used to implement checkpointing, as detailed in Appendix D.

## 4    EVALUATION

We evaluate DiffTaichi on 10 different physical simulators covering large-scale continuum and small-scale rigid body simulations. All results can be reproduced with the provided script. The dynamic/optimization processes are visualized in the supplemental video. In this section we focus our discussions on three simulators. More details on the simulators are in Appendix E.

### 4.1    DIFFERENTIABLE CONTINUUM MECHANICS FOR ELASTIC OBJECTS [diffmpm]

First, we build a differentiable continuum simulation for soft robotics applications. The physical system is governed by momentum and mass conservation, i.e. $\rho\frac{D\mathbf{v}}{Dt} = \nabla \cdot \sigma + \rho\mathbf{g}, \frac{D\rho}{Dt} + \rho\nabla \cdot \mathbf{v} = 0$. We follow ChainQueen's implementation (Hu et al., 2019b) and use the moving least squares material point method (Hu et al., 2018) to simulate the system. We were able to easily translate the original CUDA simulator into DiffTaichi syntax. Using this simulator and an open-loop controller, we can easily train a soft robot to move forward (Fig. 1, `diffmpm`).

**Performance and Productivity**   Compared with manual gradient implementations in (Hu et al., 2019b), getting gradients in DiffTaichi is effortless. As a result, the DiffTaichi implementation is $4.2\times$ shorter in terms of lines of code, and runs almost as fast; compared with TensorFlow, DiffTaichi code is $1.7\times$ shorter and $188\times$ faster (Table 1). The Tensorflow implementation is verbose due to the heavy use of `tf.gather_nd/scatter_nd` and array transposing and broadcasting.

Table 1: `diffmpm` performance comparison on an NVIDIA GTX 1080 Ti GPU. We benchmark in 2D using 6.4K particles. For the lines of code, we only include the essential implementation, excluding boilerplate code. [Reproduce: **python3 diffmpm_benchmark.py**]

| Approach | Forward Time | Backward Time | Total Time | # Lines of Code |
|---|---|---|---|---|
| TensorFlow | 13.20 ms | 35.70 ms | 48.90 ms ($188.\times$) | 190 |
| CUDA | 0.10 ms | 0.14 ms | 0.24 ms ($0.92\times$) | 460 |
| DiffTaichi | 0.11 ms | 0.15 ms | 0.26 ms ($1.00\times$) | 110 |

### 4.2    DIFFERENTIABLE INCOMPRESSIBLE FLUID SIMULATOR [smoke]

We implemented a smoke simulator (Fig. 1, `smoke`) with semi-Lagrangian advection (Stam, 1999) and implicit pressure projection, following the example in Autograd (Maclaurin et al., 2015). Using gradient descent optimization on the initial velocity field, we are able to find a velocity field that changes the pattern of the fluid to a target image (Fig. 7a in Appendix). We compare the performance of our system against PyTorch, Autograd, and JAX in Table 2. Note that as an example from the

Table 2: `smoke` benchmark against Autograd, PyTorch, and JAX. We used a $110 \times 110$ grid and 100 time steps, each with 6 Jacobi pressure projections. [Reproduce: **python3 smoke_[autograd/pytorch/jax/taichi_cpu/taichi_gpu].py**]. Note that the Autograd program uses float64 precision, which approximately doubles the run time.

| Approach | Forward Time | Backward Time | Total Time | # Essential LoC |
|---|---|---|---|---|
| PyTorch (CPU, f32) | 405 ms | 328 ms | 733 ms (13.8×) | 74 |
| PyTorch (GPU, f32) | 254 ms | 457 ms | 711 ms (13.4×) | 74 |
| Autograd (CPU, f64) | 307 ms | 1197 ms | 1504 ms (28.4×) | 51 |
| JAX (GPU, f32) | 24 ms | 75 ms | 99 ms (1.9×) | 90 |
| DiffTaichi (CPU, f32) | 66 ms | 132 ms | 198 ms (3.7×) | 75 |
| DiffTaichi (GPU, f32) | 24 ms | 29 ms | 53 ms (1.0×) | 75 |

Autograd library, this grid-based simulator is intentionally simplified to suit traditional array-based programs. For example, a periodic boundary condition is used so that Autograd can represent it using `numpy.roll`, without any branching. Still, Taichi delivers higher performance than these array-based systems. The whole program takes 10 seconds to run in DiffTaichi on a GPU, and 2 seconds are spent on JIT. JAX JIT compilation takes 2 minutes.

### 4.3 DIFFERENTIABLE RIGID BODY SIMULATORS [`rigid_body`]

We built an impulse-based (Catto, 2009) differentiable rigid body simulator (Fig. 1, `rigid_body`) for optimizing robot controllers. This simulator supports rigid body collision and friction, spring forces, joints, and actuation. The simulation is end-to-end differentiable except for a countable number of discontinuities. Interestingly, although the forward simulator works well, naively differentiating it with DiffTaichi leads to completely misleading gradients, due to the rigid body collisions. We discuss the cause and solution of this issue below.

**Improving collision gradients**  Consider the rigid ball example in Fig. 4 (left), where a rigid ball collides with a friction-less ground. Gravity is ignored, and due to conservation of kinetic energy the ball keeps a constant speed even after this elastic collision.

In the forward simulation, using a small $\Delta t$ often leads to a reasonable result, as done in many physics simulators. Lowering the initial ball height will increase the final ball height, since there is less distance to travel before the ball hits the ground and more after (see the loss curves in Fig.4, middle right). However, using a naive time integrator, no matter how small $\Delta t$ is, the evaluated gradient of final height w.r.t. initial height will be 1 instead of $-1$. This counter-intuitive behavior is due to the fact that time discretization itself is not differentiated by the compiler. Fig. 4 explains this effect in greater detail.

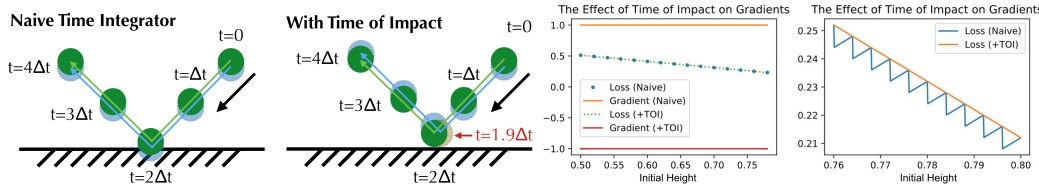

Figure 4: How gradients can go wrong with naive time integrators. For clarity we use a large $\Delta t$ here. **Left:** Since collision detection only happens at multiples of $\Delta t$ ($2\Delta t$ in this case), lowering the initial position of the ball (light blue) leads to a lowered final position. **Middle Left:** By improving the time integrator to support continuous time of impact (TOI), collisions can be detected at any time, e.g. $1.9\Delta t$ (light red). Now the blue ball ends up higher than the green one. **Middle Right:** Although the two time integration techniques lead to almost identical forward results (in practice $\Delta t$ is small), the naive time integrator gives an incorrect gradient of 1, but adding TOI yields the correct gradient. Please see our supplemental video for a better demonstration. [Reproduce: **python3 rigid_body_toi.py**] **Right:** When zooming in, the loss of the naive integrator is decreasing, and the saw-tooth pattern explains the positive gradients. [Reproduce: **python3 rigid_body_toi.py zoom**]

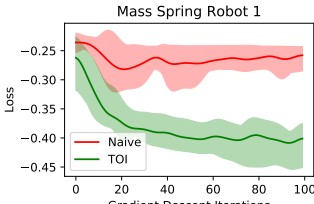 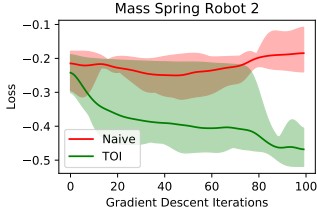 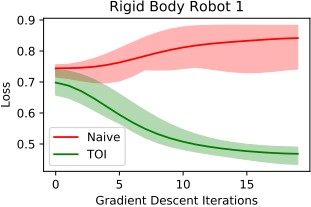

Figure 5: Adding TOI greatly improves gradient and optimization quality. Each experiment is repeated five times. [Reproduce: **python3 [mass_spring/rigid_body.py] [1/2] plot && python3 plot_losses.py**]

We propose a simple solution of adding continuous collision resolution (see, for example, Redon et al. (2002)), which considers precise time of impact (TOI), to the forward program (Fig. 4, middle left). Although it barely improves the forward simulation (Fig. 4, middle right), the gradient will be corrected effectively (Fig. 4, right). The details of continuous collision detection are in Appendix F. In real-world simulators, we find the TOI technique leads to significant improvement in gradient quality in controller optimization tasks (Fig. 5). Having TOI or not barely affects forward simulation: in the supplemental video, we show that a robot controller optimized in a simulator with TOI, actually works well in a simulator without TOI.

The takeaway is, *differentiating physical simulators does not always yield useful gradients of the physical system being simulated, even if the simulator does forward simulation well*. In Appendix G, we discuss some additional gradient issues we have encountered.

## 5 RELATED WORK

**Differentiable programming** The recent rise of deep learning has motivated the development of differentiable programming libraries for deep NNs, most notably auto-differentiation frameworks such as Theano (Bergstra et al., 2010), TensorFlow (Abadi et al., 2016) and PyTorch (Paszke et al., 2017). However, physical simulation requires complex and customizable operations due to the intrinsic computational irregularity. Using the aforementioned frameworks, programmers have to compose these coarse-grained basic operations into desired complex operations. Doing so often leads to unsatisfactory performance.

Earlier work on automatic differentiation focuses on transforming existing scalar code to obtain derivatives (e.g. Utke et al. (2008), Hascoet & Pascual (2013), Pearlmutter & Siskind (2008)). A recent trend has emerged for modern programming languages to support differentiable function transformations through annotation (e.g. Innes et al. (2019), Wei et al. (2019)). These frameworks enable differentiating general programming languages, yet they provide limited parallelism.

Differentiable array programming languages such as Halide (Ragan-Kelley et al., 2013; Li et al., 2018b), Autograd (Maclaurin et al., 2015), JAX (Bradbury et al., 2018), and Enoki (Jakob, 2019) operate on arrays instead of scalars to utilize parallelism. Instead of operating on arrays that are immutable, DiffTaichi uses an imperative style with flexible indexing to make porting existing physical simulation algorithms easier.

**Differentiable Physical Simulators** Building differentiable simulators for robotics and machine learning has recently increased in popularity. Without differentiable programming, Battaglia et al. (2016), Chang et al. (2016) and Mrowca et al. (2018) used NNs to approximate the physical process and used the NN gradients as the approximate simulation gradients. Degrave et al. (2016) and de Avila Belbute-Peres et al. (2018b) used Theano and PyTorch respectively to build differentiable rigid body simulators. Schenck & Fox (2018) differentiates position-based fluid using custom CUDA kernels. Popović et al. (2000) used a differentiable rigid body simulator for manipulating physically based animations. The ChainQueen differentiable elastic object simulator (Hu et al., 2019b) implements forward and gradient versions of continuum mechanics in hand-written CUDA kernels, leading to performance that is two orders of magnitude higher than a pure TensorFlow implementation. Liang et al. (2019) built a differentiable cloth simulator for material estimation and motion control. The deep learning community also often incorporates differentiable rendering

operations (OpenDR (Loper & Black, 2014), N3MR (Kato et al., 2018), redner (Li et al., 2018a), Mitsuba 2 (Nimier-David et al., 2019)) to learn from 3D scenes.

# 6 CONCLUSION

We have presented DiffTaichi, a new differentiable programming language designed specifically for building high-performance differentiable physical simulators. Motivated by the need for supporting megakernels, imperative programming, and flexible indexing, we developed a tailored two-scale automatic differentiation system. We used DiffTaichi to build 10 simulators and integrated them into deep neural networks, which proved the performance and productivity of DiffTaichi over existing systems. We hope our programming language can greatly lower the barrier of future research on differentiable physical simulation in the machine learning and robotics communities.

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

## A    COMPARISON WITH EXISTING SYSTEMS

**Workload differences between deep learning and differentiable physical simulation**    Existing differentiable programming tools for deep learning are typically centered around large data blobs. For example, in AlexNet, the second convolution layer has size $27 \times 27 \times 128 \times 128$. These tools usually provide users with both low-level operations such as tensor add and mul, and high-level operations such as convolution. The bottleneck of typical deep-learning-based computer vision tasks are convolutions, so the provided high-level operations, with very high arithmetic intensity[2], can fully exploit hardware capability. However, the provided operations are "atoms" of these differentiable programming tools, and cannot be further customized. Users often have to use low-level operations to compose their desired high-level operations. This introduces a lot of temporary buffers, and potentially excessive GPU kernel launches. As shown in Hu et al. (2019b), a pure TensorFlow implementation of a complex physical simulator is $132\times$ slower than a CUDA implementation, due to excessive GPU kernel launches and the lack of producer-consumer locality[3].

The table below compares DiffTaichi with existing tools for build differentiable physical simulators.

Table 3: Comparisons between DiffTaichi and other differentiable programming tools. **Note that this table only discusses features related to differentiable physical simulation**, and the other tools may not have been designed for this purpose. For example, PyTorch and TensorFlow are designed for classical deep learning tasks and have proven successful in their target domains. Also note that the XLA backend of TensorFlow and JIT feature of PyTorch allow them to fuse operators to some extent, but for simulation we want complete operator fusion within megakernels. "Swift" AD (Wei et al., 2019) is partially implemented as of November 2019. "Julia" refers to Innes et al. (2019).

| Feature | DiffTaichi | PyTorch | TensorFlow | Enoki | JAX | Halide | Julia | Swift |
|---|---|---|---|---|---|---|---|---|
| GPU Megakernels | ✓ | Δ | Δ | ✓ | ✓ | ✓ | | |
| Imperative Scheme | ✓ | | | | ✓ | | ✓ | ✓ |
| Parallelism | ✓ | ✓ | ✓ | ✓ | ✓ | ✓ | | |
| Flexible Indexing | ✓ | | | | | | ✓ | ✓ | ✓ |

## B    DIFFERENTATING STRAIGHT-LINE TAICHI KERNELS USING SOURCE CODE TRANSFORM

**Primal and adjoint kernels**    Recall that in DiffTaichi, (primal) kernels are operators that take as input multiple tensors (e.g., $X, Y$) and output another set of tensors. Mathematically, kernel $f$ has the form

$$f(X_0, X_1, .., X_n) = Y_0, Y_1, \ldots, Y_m.$$

Kernels usually execute uniform operations on these tensors. When it comes to differentiable programming, a loss function is defined on the final output tensors. The gradients of the loss function "$L$" with respect to each tensor are stored in *adjoint tensors* and computed via *adjoint kernels*.

The adjoint tensor of (primal) tensor $X_{ijk}$ is denoted as $X^*_{ijk}$. Its entries are defined by $X^*_{ijk} = \partial L / \partial X_{ijk}$. At a high level, our automatic differentiation (AD) system transforms a *primal* kernel into its *adjoint* form. Mathematically,

**(primal)** $f(X_0, X_1, .., X_n) = Y_0, Y_1, \ldots, Y_m$

⬇ Reverse-Mode Automatic Differentiation

**(adjoint)** $f^*(X_0, X_1, .., X_n, Y^*_0, Y^*_1, \ldots, Y^*_m) = X^*_0, X^*_1, .., X^*_n.$

---

[2]FLOPs per byte loaded from/stored to main memory.

[3]The CUDA kernels in Hu et al. (2019b) have much higher arithmetic intensity compared to the TensorFlow computational graph system. In other words, when implementing in CUDA immediate results are cached in registers, while in TensorFlow they are "cached" in main memory.

**Differentiating within kernels: The "make_adjoint" pass (reverse-mode AD)** After the preprocessing passes, which flatten branching and eliminate mutable local variables, the "make_adjoint" pass transforms a forward evaluation (primal) kernel into its gradient accumulation ("adjoint") kernel. It takes straight-line code directly and operates on the hierarchical intermediate representation (IR) of Taichi[4] . Multiple outer for loops are allowed for the primal kernel. The Taichi compiler will distribute these parallel iterations onto CPU/GPU threads.

During the "make_adjoint" pass, for each SSA instruction, a local adjoint variable will be allocated for gradient contribution accumulation. The compiler will traverse the statements in reverse order, and accumulate the gradients to the corresponding adjoint local variable.

For example, a 1D array operation $y_i = \sin x_i^2$ has its IR representation as follows:

```
for i ∈ range(0, 16, step 1) do
    %1 = load x[i]
    %2 = mul %1, %1
    %3 = sin(%2)
    store y[i] = %3
end for
```

The above primal kernel will be transformed into the following adjoint kernel:

```
for i in range(0, 16, step 1) do
    // adjoint variables
    %1adj = alloca 0.0
    %2adj = alloca 0.0
    %3adj = alloca 0.0
    // original forward computation
    %1 = load x[i]
    %2 = mul %1, %1
    %3 = sin(%2)
    // reverse accumulation
    %4 = load y_adj[i]
    %3adj += %4
    %5 = cos(%2)
    %2adj += %3adj * %5
    %1adj += 2 * %1 * %2adj
    atomic add x_adj[i], %1adj
end for
```

Note that for clarity the transformed code is not strictly SSA here. The actual IR has more instructions. A following simplification pass will simplify redundant instructions generated by the AD pass.

## C    COMPLEX KERNELS

Here we demonstrated how to use complex kernels to override the automatic differentiation system. We use singular value decomposition (SVD) of $3 \times 3$ matrices ($\mathbf{M} = \mathbf{U}\mathbf{\Sigma}\mathbf{V}^*$) as an example. Fast SVD solvers used in physical simulation are often iterative, yet directly evaluate the gradient of this iterative process is likely numerically unstable. Suppose we use McAdams et al. (2011) as the forward SVD solver, and use the method in Jiang (2015) (Section 2.1.1.2) to evalute the gradients, the complex kernels are used as follows:

---

[4]Taichi uses a hierarchical static single assignment (SSA) intermediate representation (IR) as its internal program representation. The Taichi compiler applies multiple transform passes to lower and simplify the SSA IR in order to get high-performance binary code.

```
# Do Singular Value Decomposition (SVD) on n matrices
@ti.kernel
def iterative_svd(num_iterations: ti.f32):
  for i in range(n):
    input = matrix_M[i]
    for iter in range(num_iterations):
      ... iteratively solve SVD using McAdams et al. 2011 ...
    matrix_U[i] = ...
    matrix_Sigma[i] = ...
    matrix_V[i] = ...

# A custom complex kernel that wraps the iterative SVD kernel
@ti.complex_kernel
def svd_forward(num_iterations):
  iterative_svd(num_iterations)

@ti.kernel
def svd_gradient():
  for i in range(n):
    ... Implement, for example, section 2.1.1.2 of Jiang (2015) ...

# A complex kernel that is registered as the svd_forward complex kernel
@ti.complex_kernel_grad(svd_forward)
def svd_backward(num_iterations):
  # differentiave SVD
  svd_gradient()
```

## D    CHECKPOINTING

In this section we demonstrate how to use checkpointing via complex kernels. The goal of checkpointing is to use recomputation to save memory space. We demonstrate this using the `diffmpm` example, whose simulation cycle consists of particle to grid transform (`p2g`), grid boundary conditions (`grid_op`), and grid to particle transform (`g2p`). We assume the simulation has $O(n)$ time steps.

### D.1    RECOMPUTATION WITHIN TIME STEPS

A naive implementation without checkpointing allocates $O(n)$ copied of the simulation grid, which can cost a lot of memory space. Actually, if we recompute the grid states during the backward simulation time step by redoing `p2g` and `grid_op`, we can reused the grid states and allocate only one copy. This checkpointing optimization is demonstrated in the code below:

```
@ti.complex_kernel
def advance(s):
  clear_grid()
  compute_actuation(s)
  p2g(s)
  grid_op()
  g2p(s)

@ti.complex_kernel_grad(advance)
def advance_grad(s):
  clear_grid()
  p2g(s)
  grid_op() # recompute the grid

  g2p.grad(s)
  grid_op.grad()
  p2g.grad(s)
  compute_actuation.grad(s)
```

### D.2    SEGMENT-WISE RECOMPUTATION

Given a simulation with $O(n)$ time steps, if all simulation steps are recorded, the space consumption is $O(n)$. This linear space consumption is sometimes too large for high-resolution simulations with long time horizon. Fortunately, we can reduce the space consumption using a segment-wise

checkpointing trick: We split the simulation into segments of $S$ steps, and in forward simulation store only the first simulation state in each segment. During backpropagation when we need the remaining simulation states in a segment, we recompute them based on the first state in that segment.

Note that if the segment size is $O(S)$, then we only need to store $O(n/S)$ simulation steps for checkpoints and $O(S)$ reusable simulation steps for backpropagation within segments. The total space consumption is $O(S + n/S)$. Setting $S = O(\sqrt{n})$ reduces memory consumption from $O(n)$ to $O(\sqrt{n})$. The time complexity remains $O(n)$.

# E  DETAILS ON 10 DIFFERENTIABLE SIMULATORS

## E.1  DIFFERENTIABLE CONTINUUM MECHANICS FOR ELASTIC OBJECTS [diffmpm]

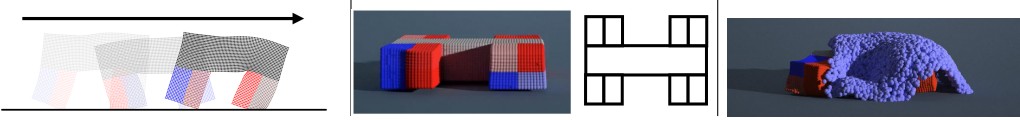

Figure 6: Controller optimization with our differentiable continuum simulators. **Left:** the 2D robot with four muscles. **Middle:** A 3D robot with 16 muscles and 30K particles crawling on the ground. [Reproduce: **python3 [diffmpm/diffmpm3d].py**] **Right:** We couple the robot (30K particles) and the liquid simulator (13K particles), and optimize its open-loop controller in this difficult situation.[Reproduce: **python3 liquid.py**]

## E.2  DIFFERENTIABLE LIQUID SIMULATOR [liquid]

We follow the weakly compressible fluid model in Tampubolon et al. (2017) and implemented a 3D differentiable liquid simulator within the [diffmpm3d] framework. Our liquid simulation can be two-way coupled with elastic object simulation (Figure 6, right).

## E.3  DIFFERENTIABLE INCOMPRESSIBLE FLUID SIMULATOR [smoke]

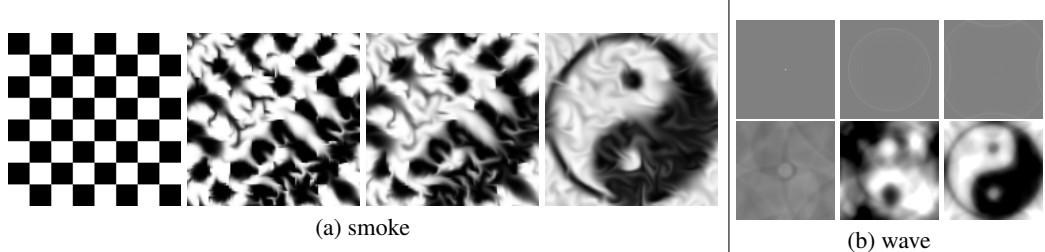

(a) smoke

(b) wave

Figure 7: (a): (Left to right) with an optimized initial smoke velocity field, the fluid changes its pattern to a "Taichi" symbol. [Reproduce: **python3 smoke_taichi.py**] (b): Unoptimized (top three) and optimized (bottom three) waves at time step 3, 189, and 255. [Reproduce: **python3 wave.py**]

**Backpropagating Through Pressure Projection**   We followed the baseline implementation in Autograd, and used 10 Jacobi iterations for pressure projection. Technically, 10 Jacobi iterations are not sufficient to make the velocity field fully divergence-free. However, in this example, it does a decent job, and we are able to successfully backpropagate through the unrolled 10 Jacobi iterations.

In larger-scale simulations, 10 Jacobi iterations are likely not sufficient. Assuming the Poisson solve is done by an iterative solver (e.g. multigrid preconditioned conjugate gradients, MGPCG) with 5 multigrid levels and 50 conjugate gradient iterations, then automatic differentiation will likely not be able to provide gradients with sufficient numerical accuracy across this long iterative process. The accuracy is likely worse when conjugate gradients present, as they are known to numerically drift as the number of iterations increases. In this case, the user can still use DiffTaichi to implement the forward MGPCG solver, while implementing the backward part of the Poisson solve manually,

likely using adjoint methods (Errico, 1997). DiffTaichi provides "complex kernels" to override the built-in AD system, as shown in Appendix C.

### E.4 DIFFERENTIABLE HEIGHT FIELD SHALLOW WATER SIMULATOR [wave]

We adopt the wave equation in Wang et al. (2018) to model shallow water height field evolution:

$$\ddot{u} = c^2\nabla^2 u + c\alpha\nabla^2\dot{u}, \tag{1}$$

where $u$ is the height of shallow water, $c$ is the "speed of sound" and $\alpha$ is a damping coefficient. We use the $\dot{u}$ and $\ddot{u}$ notations for the first and second order partial derivatives of $u$ w.r.t time $t$ respectively.

Wang et al. (2018) used the finite different time-domain (FDTD) method (Larsson & Thomée, 2008) to discretize Eqn. 1, yielding an update scheme:

$$u_{t,i,j} = 2u_{t-1,i,j} + (c^2\Delta t^2 + c\alpha\Delta t)(\nabla^2 u)_{t-1,i,j} - p_{t-2,i,j} - c\alpha\Delta t(\nabla^2 u)_{t-2,i,j},$$

where

$$(\nabla^2 u)_{t,i,j} = \frac{-4u_{t,i,j} + u_{t,i,j+1} + u_{t,i,j-1} + u_{t,i+1,j} + u_{t,i-1,j}}{\Delta x^2}.$$

We implemented this wave simulator in DiffTaichi to simulate shallow water. We used a grid of resolution $128 \times 128$ and 256 time steps. The loss function is defined as

$$L = \sum_{i,j}\Delta x^2(u_{T,i,j} - \hat{u}_{i,j})^2$$

where $T$ is the final time step, and $\hat{u}$ is the target height field. 200 gradient descent iterations are then used to optimize the initial height field. We set $\hat{u}$ to be the pattern "Taichi", and Fig. 7b shows the unoptimized and optimized wave evolution.

We set the "Taichi" symbol as the target pattern. Fig. 7b shows the unoptimized and optimized final wave patterns. More details on discretization is in Appendix E.

### E.5 DIFFERENTIABLE MASS-SPRING SYSTEM [mass_spring]

We extend the mass-spring system in the main text with ground collision and a NN controller. The time-of-impact fix is implemented for improved gradients. The optimization goal is to maximize the distance moved forward with 2048 time steps. We designed three mass-spring robots as shown in Fig. 8 (left).

### E.6 DIFFERENTIABLE BILLIARD SIMULATOR [billiards]

A differentiable rigid body simulator is built for optimizing a billiards strategy (Fig. 8, middle). We used forward Euler for the billiard ball motion and conservation of momentum and kinetic energy for collision resolution.

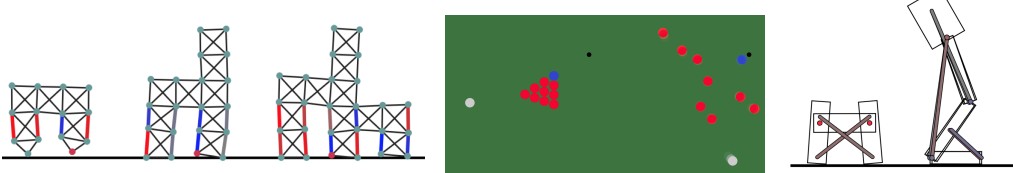

Figure 8: **Left:** Three mass-spring robots. The red and blue springs are actuated. A two layer NN is used as controller. [Reproduce: **python3 mass_spring.py [1/2/3] train**]. **Middle:** Optimizing billiards. The optimizer adjusts the initial position and velocity of the white ball, so that the blue ball will reach the target destination (black dot). [Reproduce: **python3 billiards.py**] **Right:** Optimizing a robot walking. The rigid robot is controlled with a NN controller and learned to walk in 20 gradient descent iterations. [Reproduce: **python3 rigid_body.py**]

## E.7   DIFFERENTIABLE RIGID BODY SIMULATOR [rigid_body]

**Are rigid body collisions differentiable?**   It is worth noting that discontinuities can happen in rigid body collisions, and at a countable number of discontinuities the objective function is non-differentiable. However, apart from these discontinuities, the process is still differentiable almost everywhere. The situation of rigid body collision is somewhat similar to the "ReLU" activation function in neural networks: at point $x = 0$, ReLU is not differentiable (although continuous), yet it is still widely adopted. The rigid body simulation cases are more complex than ReLU, as we have not only non-differentiable points, but also discontinuous points. Based on our experiments, in these impulse-based rigid body simulators, we still find the gradients useful for optimization despite the discontinuities, especially with our time-of-impact fix.

## E.8   DIFFERENTIABLE WATER RENDERER [water_renderer]

We implemented differentiable renderers to visualize the refracting water surfaces from wave. We use finite differences to reconstruct the water surface models based on the input height field and refract camera rays to sample the images, using bilinear interpolation for meaningful gradients. To show our system works well with other differentiable programming systems, we use an adversarial optimization goal: fool VGG-16 into thinking that the refracted squirrel image is a goldfish (Fig. 9).

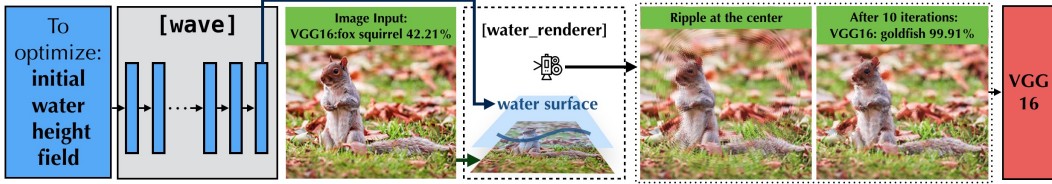

Figure 9: This three-stage program (simulation, rendering, recognition) is end-to-end differentiable. Our optimized initial water height field evolves to form a refraction pattern that perturbs the image into one that fools VGG16 (99.91% goldfish). [Reproduce: **python3 water_renderer.py**]

## E.9   DIFFERENTIABLE VOLUME RENDERER [volume_renderer]

We implemented a basic volume renderer that simply uses ray marching (we ignore light, scattering, etc.) to integrate a density field over each camera ray. In this task, we render a number of target images from different viewpoints, with the camera rotated around the given volume. The goal is then to optimize for the density field of the volume that would produce these target images: we render candidate images from the same viewpoints and compute an L2 loss between them and the target images, before performing gradient descent on the density field (Fig. 10). Essentially, this demonstrates how to use gradients to reconstruct 3D objects out of X-ray photos in a brute-force manner. Other approaches to this task include algebraic reconstruction techniques (ART)  (Gordon et al., 1970).

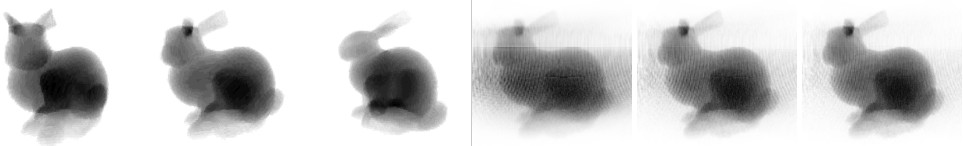

Figure 10: Volume rendering of bunny shaped density field. **Left:** 3 (of the 7) target images. **Right:** optimized images of the middle bunny after iteration 2, 50, 100. [Reproduce: **python3 volume_renderer.py**]

### E.10 DIFFERENTIABLE ELECTRIC FIELD SIMULATOR [electric]

Recall Coulomb's law: $\mathbf{F} = k\frac{q_1 q_2}{r^2}\hat{\mathbf{r}}$. In the right figure, there are eight electrodes carrying static charge. The red ball also carries static charge. The controller, which is a two-layer neural network, tries to manipulate the electrodes so that the red ball follows the path of the blue ball. The bigger the electrode, the more positive charge it carries.

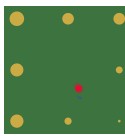

## F FIXING GRADIENTS WITH TIME OF IMPACT AND CONTINUOUS COLLISION DETECTION

Here is a naive time integrator in the mass-spring system example:

```
@ti.kernel
def advance(t: ti.i32):
  for i in range(n_objects):
    s = math.exp(-dt * damping)
    new_v = s * v[t - 1, i] + dt * gravity * ti.Vector([0.0, 1.0])
    old_x = x[t - 1, i]
    depth = old_x[1] - ground_height
    if depth < 0 and new_v[1] < 0:
      # assuming a sticky ground (infinite coefficient of friction)
      new_v[0] = 0
      new_v[1] = 0

    # Without considering time of impact, we assume the whole dt uses new_v
    new_x = old_x + dt * new_v

    v[t, i] = new_v
    x[t, i] = new_x
```

Implementing TOI in this system is relative straightforward:

```
@ti.kernel
def advance_toi(t: ti.i32):
  for i in range(n_objects):
    s = math.exp(-dt * damping)
    old_v = s * v[t - 1, i] + dt * gravity * ti.Vector([0.0, 1.0])
    old_x = x[t - 1, i]
    new_x = old_x + dt * old_v
    toi = 0.0
    new_v = old_v
    if new_x[1] < ground_height and old_v[1] < -1e-4:
      # The 1e-4 safe guard is important for numerical stability
      toi = -(old_x[1] - ground_height) / old_v[1] # Compute the time of impact
      new_v = ti.Vector([0.0, 0.0])

    # Note that with time of impact, dt is divided into two parts,
    #   the first part using old_v, and second part using new_v
    new_x = old_x + toi * old_v + (dt - toi) * new_v

    v[t, i] = new_v
    x[t, i] = new_x
```

In rigid body simulation, the implementation follows the same idea yet is slightly more complex. Please refer to `rigid_body.py` for more details.

## G ADDITIONAL TIPS ON GRADIENT BEHAVIORS

**Initialization matters: flat lands and local minima in physical processes**  A trivial example of objective flat land is in `billiards`. Without proper initialization, gradient descent will make no progress since gradients are zero (Fig. 11). Also note the local minimum near $(-5, 0.03)$.

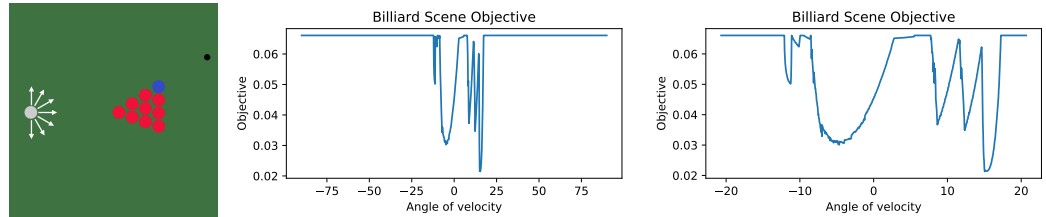

Figure 11: **Left:** Scanning initial velocity in the billiard example. **Middle:** Most initial angles yield a flat objective (final distance between the blue ball and black destination) of $0.065$, since the white ball does not collide with any other balls and imposes no effect on the pink ball via the chain reaction. **Right:** A zoomed-in view of the middle figure. The complex collisions lead to a lot of local minimums. [Reproduce: **python3 billiards.py 1.0/0.23**]

In `mass_spring` and `rigid_body`, once the robot falls down, gradient descent will quickly become trapped. A robot on the ground will make no further progress, no matter how it changes its controller. This leads to a more non-trivial local minimum and zero gradient case.

**Ideal physical models are only "ideal": discontinuities and singularities**   Real-world macroscopic physical processes are usually continuous. However, building upon ideal physical models, even in the forward physical simulation results can contain discontinuities. For example, in a rigid body model with friction, changing the initial rotation of the box can lead to different corners hitting the ground first, and result in a discontinuity (Fig. 12). In `electric` and `mass_spring`, due to the $\frac{1}{r^2}$ and $\frac{1}{r}$ terms, when $r \to 0$, gradients can be very inaccurate due to numerical precision issues. Note that $d(1/r)/dr = -1/r^2$, and the gradient is more numerically problematic than the primal for a small $r$. Safeguarding $r$ is critically important for gradient stability.

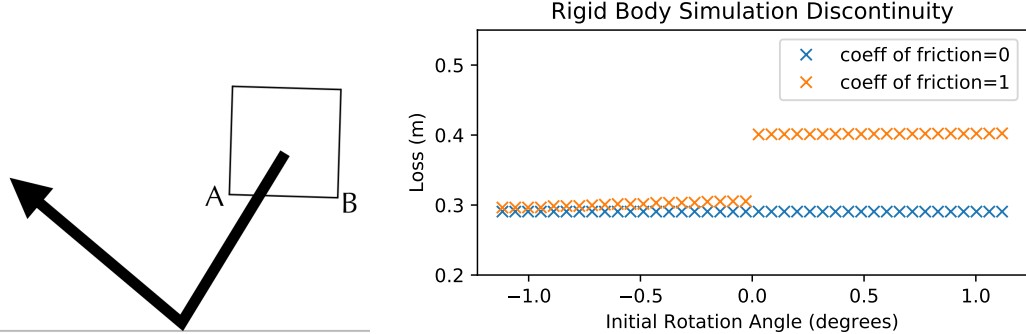

Figure 12: Friction in rigid body with collision is a common source of discontinuity. In this scene a rigid body hits the ground. Slightly rotating the rigid body changes which corner (A/B) hits the ground first, and different normal/friction impulses will be applied to the rigid body. This leads to a discontinuity in its final position (loss=final y coordinate). [Reproduce: **python3 rigid_body_discontinuity.py**] Please see our supplemental video for more details.

