# OpenReview forum: "DiffTaichi: Differentiable Programming for Physical Simulation"
_ICLR.cc/2020/Conference — Accept (Poster)_

### Official Review · AnonReviewer1 · 2019-10-20
**Official Blind Review #1**

**Rating:** 6

**Review:**

This paper introduces DiffSim, a programming language for high-performance differentiable physics simulations. The paper demonstrates 10 different simulations with controller optimization. It shows that the proposed language is easier to use and faster than the other alternatives, such as CUDA and TensorFlow. At the end, the paper provides insightful discussions why the gradient of the simulation could be wrong.

Differentiable physics simulation is an important research area, especially for optimal control and reinforcement learning. While I am impressed by the large variety of examples demonstrated in the paper, I am leaning towards rejecting the paper because of its poor presentation. The paper only gives a simple and high-level example of the language (optimizing the rest length of springs that form a triangle), very brief descriptions of 10 examples and some discussions about the difficulty of computing useful gradients, but without any in-depth discussion how everything is implemented. This is not enough for an ICLR paper. For example, the paper does not answer some of the fundamental problems of differentiable physics. For example, collision and contact are inherently non-differentiable. How does the paper handle it in the examples of locomotion and billiards (Figure 4)? In addition, how does the paper back-propagate the gradient through the incompressibility conditions (Poisson solve) of fluid simulation?

Here is my suggestions how to improve the writing. There are several ways to write the paper, with different emphasis. If this paper is more about introducing a new programming language, Appendix B Compiler Design and Implementation would be important and should be moved to main text. If the paper want to emphasize how to handle the non-differentiable cases of the simulation, then detailed derivations of contact, collision, and linear/nonlinear solving (due to incompressibility conditions or implicit integrators) should be presented. If the paper would like to demonstrate how differentiable physics simulation can help with controller optimization, then two to three examples, such as the locomotion control for soft bodies or rigid bodies, should be analyzed in far more details, and compared with traditional method without differentiable simulation. It is good to focus on one of the above points, based on the venue that this paper is submitted to. Currently, the paper is trying to touch all three. But due to the page limit, it is not thorough, or detailed in any one of them.

-------------------------Update after rebuttal------------------------------
Thank you for the revision of the paper and the additional comparisons with Jax. The revised version reads much better. The response and the revision addressed most of my concerns. Thus, I raised my rating to weak accept.

**Experience Assessment:**

I have read many papers in this area.

**Review Assessment: Checking Correctness Of Derivations And Theory:**

I assessed the sensibility of the derivations and theory.

**Review Assessment: Checking Correctness Of Experiments:**

I assessed the sensibility of the experiments.

**Review Assessment: Thoroughness In Paper Reading:**

I read the paper at least twice and used my best judgement in assessing the paper.

---

> ### Author Response · Authors · 2019-11-10
> **Response to Reviewer 1**
>
> Dear Reviewer 1,
>
> Thank you very much for the helpful writing suggestions. We have adopted your writing strategy. Now the presentation is focused on the DiffSim system itself, i.e. how to build an efficient and easy-to-use automatic differentiation system for physical simulation.
>
> We will add details of every physical simulator in the appendix in the next update, as it may take a while to document 10 different differentiable simulators. Here we briefly answer your questions:
>
> ** Rigid Body Collision Gradients **
> It is true that discontinuities can happen in rigid body collisions, and at a countable number of discontinuities the objective function is nondifferentiable. However, apart from these discontinuities, the process is still differentiable almost everywhere. The situation of rigid body collision is somewhat similar to the “ReLU” activation function in neural networks: at point x=0, ReLU is not differentiable (although continuous), yet it is still widely adopted. The rigid body simulation cases are more complex than ReLU, as we have not only non-differentiable points, but also discontinuous points. Based on our experiments, in these impulse-based rigid body simulators (rigid_body and billiards), we still find the gradients useful for optimization, despite the discontinuities, especially with our time-of-impact fix.
>
>
> ** Pressure Projection Gradients in Incompressible Fluids **
> We followed the baseline implementation in Autograd, and used 10 Jacobi iterations for pressure projection. Technically, 10 Jacobi iterations are not sufficient to make the velocity field fully divergence-free. However, in this example, it does a decent job, and we are able to successfully backpropagate through the unrolled 10 Jacobi iterations.
>
> In larger-scale fluid simulations, 10 Jacobi iterations are likely not sufficient. Assuming the Poisson solve is done by an iterative solver (e.g. multigrid preconditioned conjugate gradients, MGPCG) with 5 multigrid levels and 50 conjugate gradient iterations, then automatic differentiation will likely not be able to provide gradients with sufficient numerical accuracy across this long iterative process. The accuracy is likely worse when conjugate gradients present, as they are known to numerically drift as the number of iterations increases. In this case, the user can still use DiffSim to implement the forward MGPCG solver, while implementing the backward part of the Poisson solve manually, likely using adjoint methods [1]. DiffSim provides “complex kernels” to override the built-in AD system, as shown in appendix C.
>
>
> We have included the above discussions in Appendix E to help future readers, although now we focus more on the system perspective in the main paper.
>
> Please also find the detailed paper change log in our general response to all reviewers. Thank you again for your time and very constructive feedback!
>
> Best,
> Authors
>
> [1]``What is an adjoint model?’’ by Ronald M Errico, in Bulletin of the American Meteorological Society, 1997

---

### Official Review · AnonReviewer2 · 2019-10-22
**Official Blind Review #2**

**Rating:** 3

**Review:**

This paper presents a programming language for building differentiable physics simulators. This is a very interesting goal, as differentiable systems are a crucial building block for many deep learning methods and similar optimization techniques.

The system presented by the authors is certainly impressive. Unfortunately, the paper itself covers a wide range of topics, and consists of an overview of the language with a programming tutorial, a collection of ten results, and a brief discussion of problems when computing gradients.

The core of the proposed work, the programming language seems to be quite powerful. However, it seems to be built on an existing system, which was published as a programming language for simulation in this years siggraph asia conference (Taichi: A language for high-performance computation on spatially sparse data structures. In SIGGRAPH Asia 2019 Technical Papers, pp. 201. ACM, 2019a). This ICLR submission seems to extend this system to build and provide gradient information automatically along with the simulation itself. There seem to be few technical challenges here, and many aspect discussed in section 2 are shared with the original simulation language.

The examples cover a nice range of cases, from simple mass spring systems and a rendering case to complex 3d simulations. Here, I was a bit surprised that the paper only compares to autograd, which has been succeeded by jax. The latter also provides a compiler backend to produce GPU code with gradients, and as such seems very closely related to the proposed language. From the submission, it's hard to say which version has advantages. The examples seem to be a sequence of demos of the language, rather than illustrating different technical challenges or improvements for a scientific conference. Or at least a discussion of these differences is currently missing in the text.

Section four also mostly gives the impression of a loose discussion. The gradients for rigid body impacts are interesting, but seem relevant only for a subset of 2D examples shown in the paper. The discussion of gradient explosions is quite ad-hoc, and would be stronger with a more detailed analysis.

The submission as a whole aims for a very interesting direction, but I think the paper would benefit from focusing on a certain range of problems, such as the rigid body control cases, in conjunction with topics such as the improved gradients. Instead, the current version tries to combine this topic with a systems overview, a tutorial and loosely related discussions. Combined with the length of 10 pages, I think the work could use a revision rather than being accepted in its current form.

**Experience Assessment:**

I have published one or two papers in this area.

**Review Assessment: Checking Correctness Of Derivations And Theory:**

I assessed the sensibility of the derivations and theory.

**Review Assessment: Checking Correctness Of Experiments:**

I carefully checked the experiments.

**Review Assessment: Thoroughness In Paper Reading:**

I read the paper thoroughly.

---

> ### Author Response · Authors · 2019-11-10
> **Response to Reviewer 2**
>
> Dear Reviewer 2,
>
> Thank you for the helpful feedback. We have reorganized the paper to make it more focused on the DiffSim automatic differentiation system design alone, instead of multiple topics. The paper is now 8 pages instead of 10.
>
> ** Building on Top of Taichi**
> The main goal of DiffSim is to simplify the process of making existing physical simulators differentiable, and we reuse Taichi because Taichi is very suitable for building forward simulators. We indeed reused some infrastructure of Taichi, but such reuse also poses a unique challenge to redesign a tailored automatic differentiation system for it, which a) does not harm the performance of Taichi programs and b) needs minimal code modification to make a Taichi program differentiable. To this end, we have developed a tailored two-scale AD system that is high-performance and imposes minimal global data access restrictions to Taichi programs. Please check out the new section 3 for more details.
>
> **Comparison with JAX**
> We have added JAX with GPU backend to the smoke simulation benchmark, and DiffSim GPU is 1.9x faster than JAX GPU, despite that this grid-based simulation benchmark is slightly biased towards differentiable array programming systems such as Autograd and JAX. The whole program takes 10 seconds to run in DiffSim on a GPU, and 2 seconds are spent on JIT. JAX JIT compilation takes 2 minutes. In Appendix A, we also added a table that comprehensively compares DiffSim with 7 other existing systems.
>
> **Gradient Quality Discussions**
> We removed the discussion on gradient explosion, and moved the rigid body gradient issue to the evaluation section along with the rigid body simulator, to limit the scope of this gradient discussion to rigid body simulations.
>
> Please also find the detailed paper change log in our general response to all reviewers. Thank you again for your time and feedback!
>
> Best,
> Authors

---

### Official Review · AnonReviewer3 · 2019-11-02
**Official Blind Review #3**

**Rating:** 6

**Review:**

*Summary*
This paper describes DiffSim, a differentiable programming system for learning with physical simulation. The system (built on the Taichi system) allows users to specify a forward simulation in a Python-like syntax, after which the program is compiled and iteratively run in both forward-mode and gradients computed for system parameters and controllers, as desired. A variety of simple simulations are included, demonstrating that the automatically generated CUDA code runs as fast as hand-written CUDA code (and noticeably faster than TensorFlow or PyTorch implementations), while requiring far fewer lines of code. The final section details two issues--time of  impact errors due to discrete time intervals and gradient explosions with long time horizons--and some potential solutions.

*Rating*
The paper is interesting and easy to read. While some part of the underlying functionality of DiffSim is directly derived from previous work (Taichi), the paper does describe a non-trivial contribution.

I lack the background to comment constructively about expectations for these simulations or the fidelity of the methods in this paper. What evidence can you offer regarding the physical fidelity achievable and how that relates to issues of scalability, gradient behavior, size of time steps, code complexity, etc.? For a sense of context, what might be needed to simulate a 7 DoF robotic arm and learn a controller that would reasonably transfer to a real robot?

Overall, I'm optimistic about this paper, and would tend to vote for acceptance.

*Notes*
pg3: define k (spring stiffness?)
pg4: what is the value of 'mass' for this simulation?
Fig 8: what is the x-axis in the two right plots? initial height?
Fig 10: right plot title should probably be "Gradient Explosion with Damping"

**Experience Assessment:**

I do not know much about this area.

**Review Assessment: Checking Correctness Of Derivations And Theory:**

I assessed the sensibility of the derivations and theory.

**Review Assessment: Checking Correctness Of Experiments:**

I assessed the sensibility of the experiments.

**Review Assessment: Thoroughness In Paper Reading:**

I made a quick assessment of this paper.

---

> ### Author Response · Authors · 2019-11-10
> **Response to Reviewer 3**
>
> Dear Reviewer 3,
>
> Thank you for the positive feedback. The question about simulation fidelity is very interesting. DiffSim is as expressive as traditional languages such as C++/Fortran in building physical simulators, so it can achieve the fidelity level of previously build simulators. In order to simulate a 7-DoF robotic arm, a differentiable rigid body simulator written in DiffSim should be used to train the controller, but transferring the controller to a real robot would face a sim2real gap, just as physical simulators written in any other language. DiffSim does not directly address this gap, but it does significantly reduce code complexity and would allow researchers to develop more realistic simulators with the same amount of work. Similarly, DiffSim does not resolve the numerical accuracy issue caused by a finite time step size, but it does improve the program performance to allow users to run simulations with smaller time step sizes and higher spatial resolution and thereby smaller discretization errors.
>
> We have also fixed the minor issues in the revision:
>    -   (Page 3) k is spring stiffness.
>    -   (Page 4) We used mass = 1 throughout the mass-spring simulation.
>    -   (Fig. 8) The x-axes are initial height.
>    -   (Fig. 10) Thanks for pointing out the typo in “Gradient Explosion with Damping”. As suggested by reviewer 2, we have removed the discussion on gradient explosion.
>
> Please also find the detailed paper change log in our general response to all reviewers. Thank you again for your time and feedback.
>
> Best,
> Authors

---

### Author Response · Authors · 2019-11-10
**General Response to All Reviewers: Paper Reorganized**

Dear Reviewers,

Thank you so much for the constructive feedback! We have reorganized the paper according to your suggestions. The paper now focuses on the DiffSim system itself, i.e. how to build an efficient and easy-to-use automatic differentiation system for physical simulation. The discussion on gradient behavior is demoted to be part of system evaluation. The paper is now 8 pages instead of 10. Pdf link: https://openreview.net/pdf?id=B1eB5xSFvr

Structural Reorganization:
   -   The introduction is updated to suit the new paper structure.
   -   Moved the background on Taichi from Appendix to the main text as section 2.
   -   Added section 3 (Automatically Differentiate Physical Simulation), which details automatic differentiation on Taichi. Two key components are local automatic differentiation within kernels, and global AD across kernels using a light-weight tape. Part of the old Appendix B (Compiler Design and Implementation) is now promoted to this section.
   -   Section 4 (Evaluation) now focuses on only 3 examples instead of 10. The remaining 7 examples are moved to Appendix. We will also include the implementation details of all the simulators in Appendix in a later update.
   -   The old section 4 (Robust Gradients) is significantly shortened and merged into the new section 4. Discussions of the gradient explosion issue are now removed. The only discussion of gradients left in the main text is rigid body collision gradients.

Minor Changes:
   -   As suggested by R2, we have made a comparison with JAX as well. We reran the whole smoke benchmark with 6 Jacobi iterations (used to be 10), since JAX crashes when we use 7 or more iterations. Also, we have improved the DiffSim code generation and therefore its performance is improved.
   -   Fixed typos as pointed out by R3.
   -   In Appendix, we added a table that comprehensively compares DiffSim with 7 other existing systems. We also added examples of complex kernels (Appendix C) and checkpointing (Appendix D).

We have addressed comments from every reviewer in separate replies. Please also check that out. Thank you again for your suggestions, and we are happy to further improve the paper if you have any new advice! :-)

Best,
Authors

---

### Decision · Program_Chairs · 2019-12-19

**Decision:**

Accept (Poster)

**Comment:**

The paper provides a language for optimizing through physical simulations. The reviewers had a number of concerns related to paper organization and insufficient comparisons to related work (jax). During the discussion phase, the authors significantly updated their paper and ran additional experiments, leading to a much stronger paper.